

# Neoadjuvant chemotherapy efficacy and prognosis in HER2-low and HER2-zero breast cancer patients by HR status: a retrospective study in China

Shaorong Zhao[1,2,3,4,*], Yuyun Wang[1,2,3,4,*], Angxiao Zhou[1,2,3,4], Xu Liu[1,2,3,4], Yi Zhang[1,2,3,4] and Jin Zhang[1,2,3,4]

[1] The Third Department of Breast Cancer, Tianjin Medical University Cancer Institute and Hospital, National Clinical Research Center for Cancer, Tianjin, China
[2] Tianjin's Clinical Research Center for Cancer, Tianjin Medical University Cancer Institute and Hospital, Tianjin, China
[3] Key Laboratory of Breast Cancer Prevention and Therapy, Tianjin Medical University, Ministry of Education, Tianjin, China
[4] Key Laboratory of Cancer Prevention and Therapy, Tianjin Medical University Cancer Institute and Hospital, Tianjin, China
* These authors contributed equally to this work.

Corresponding author
Jin Zhang,
zhangjintjmuch1@163.com

## ABSTRACT

**Background:** The promising efficacy of novel anti-HER2 antibody-drug conjugates (ADC) in HER2-low breast cancer has made HER2-low a research hotspot. However, controversy remains regarding the neoadjuvant chemotherapy (NAC) efficacy, prognosis, and the relationship with hormone receptor (HR) status of HER2-low.
**Methods:** A retrospective analysis was conducted on 975 patients with HER2-negative breast cancer undergoing NAC at Tianjin Medical University Cancer Institute and Hospital, evaluating pathological complete response (pCR) rate and prognosis between HER2-low and HER2-zero in the overall cohort and subgroups.
**Results:** Overall, 579 (59.4%) and 396 (40.6%) patients were HER2-low and HER2-zero disease, respectively. Compared with HER2-zero, the HER2-low cohort consists of more postmenopausal patients, with lower histological grade and higher HR positivity. In the HR-positive subgroup, HER2-low cases remain to exhibit lower histological grade, while in the HR-negative subgroup, they show higher grade. The HER2-low group had lower pCR rates than the HER2-zero group (16.4% *vs.* 24.0%). In the HR-positive subgroup, HER2-low consistently showed lower pCR rate (8.1% *vs.* 15.5%), and served as an independent suppressive factor for the pCR rate. However, no significant difference was observed in the pCR rates between HER2-low and HER2-zero in the HR-negative breast cancer. In the entire cohort and in stratified subgroups based on HR and pCR statuses, no difference in disease-free survival were observed between HER2-low and HER2-zero.
**Conclusions:** In the Chinese population, HER2-low breast cancer exhibits distinct characteristics and efficacy of NAC in different HR subgroups. Its reduced pCR rate in HR-positive subgroup is particularly important for clinical decisions. However, HER2-low is not a reliable factor for assessing long-term survival outcomes.

# INTRODUCTION

Breast cancer is a prevalent malignancy in women with high incidence and mortality rates. A total of 2.26 million new diagnoses were reported in 2020, leading to approximately 680,000 deaths (*Sung et al., 2021*). Breast cancer is categorized into four types, namely Luminal A, Luminal B, HER2-positive and triple-negative breast cancer (TNBC), and their treatment strategies and prognoses differ significantly (*Gradishar et al., 2022*).

The human epidermal growth factor receptor 2 (HER2) gene is considered as an important indicator of tumor type, treatment decision, and disease prognosis. HER2 is a transmembrane glycoprotein epidermal growth factor receptor (EGFR) with tyrosine kinase activity encoded by the ERBB2 gene. Breast cancer with high HER2 expression is more aggressive and exhibit worse prognosis, and is recognized as a distinct biological subtype. The anti-HER2 treatments have been developed and have significantly improved the treatment outcomes for patients with high HER2 expression (*Piccart et al., 2021*; *Swain et al., 2020*). According to the 2018 guidelines from the American Society of Clinical Oncology and the College of American Pathologists (ASCO/CAP), HER2-positive breast cancer is characterized by overexpression of HER2 in immunohistochemistry (IHC) testing score of 3+ and/or gene amplification in fluorescent *in situ* hybridization (FISH) testing. If the HER2 IHC score is 0, 1+, or 2+ and FISH is negative, the breast cancer is categorized as HER2-negative. HER2-positive status is a strong predictor of sensitivity to HER2-targeted therapies, whereas HER2-negative BC is not recommended for anti-HER2 treatments (*Wolff et al., 2018*). HER2-negative cases can be either Luminal or TNBC and are treated accordingly; however, this is both imprecise and inadequate.

The results of recent clinical trials indicate that some patients with breast cancer previously categorized as HER2-negative, who have low HER2 expression with IHC 1+ or IHC 2+ and negative FISH results (IHC 2+/FISH−), can benefit from novel anti-HER2 antibody-drug conjugates (ADCs) (*Banerji et al., 2019*; *Modi et al., 2020*; *Tarantino et al., 2020*; *van der Lee et al., 2015*). However, limited information exists on clinicopathological features, responses to chemotherapy regimens, and outcome prediction in HER2-low patients (*Bao et al., 2021*; *de Nonneville et al., 2022*; *Tarantino et al., 2022*; *Zhang et al., 2022a*), especially in the patients receiving neoadjuvant chemotherapy (NAC). Preoperative or neoadjuvant therapy for breast cancer includes NAC, targeted therapy, endocrine therapy, and even immunotherapy. NAC is currently one of the most widely employed standard treatments in breast cancer. It generally utilized to transform locally advanced breast tumors into operable ones. For most operable tumors, downstaging with NAC can increase the rate of breast conservation (from 7% to 12%) (*Schott & Hayes, 2012*). In addition, considering the efficacy of NAC, particularly in achieving pathologic complete response (pCR), it can serve as an indicator of tumor drug sensitivity (*Wang & Mao, 2020*). However, controversy remains regarding the efficacy of NAC for the HER2-low

population and their long-term survival outcomes post-NAC (*Ma et al., 2023*; *Yang et al., 2023*; *Zhou et al., 2023*).

Thus, this study aimed to evaluate the pCR and disease-free survival (DFS) among patients with HER2-low expression undergoing NAC. We believe that this study is relevant for both the treatment and prognosis of HER2-low patients.

## MATERIALS AND METHODS

### Collection of patient data

The data of 975 patients with breast cancer who underwent NAC at Tianjin Medical University Cancer Institute and Hospital between January 2014 and December 2017 were retrospectively analyzed. The data included information on age; menopausal status; clinical stage (T and N); pathological type; histological grade; hormone receptor (HR); estrogen receptor (ER); progesterone receptor (PR); HER2 status; Ki-67; and NAC regimens and efficacy.

### Sample size

This study only calculated the sample size for the primary outcome of pathological complete response (pCR) rate using an online sample size calculator (https://sample-size.net/sample-size-proportions/). Sample size was calculated using $\alpha = 0.05$ and power 80%. The output parameters included: sample size Group 1 = 236 and Group 0 = 339. In this study, a total of 975 patients were analyzed, with 396 in the HER2-zero group and 579 in the HER2-low group, exceeding the minimum sample requirement.

### Inclusion and exclusion criteria and study design

Patients with (1) histologically confirmed invasive breast cancer by thick needle biopsy, (2) NAC before surgical resection (modified radical mastectomy or breast conservation), pathology examination after surgery, and (3) available comprehensive clinical information were included. Patients with (1) bilateral, inflammatory, or occult breast carcinoma; (2) stage IV disease; (3) presence of other malignancies; (4) concurrently receiving other breast cancer related treatments; and (5) discontinued NAC due to intolerance were excluded. The study was approved by Medical Ethics Committee of Tianjin Medical University Cancer Institute and Hospital (bc2023009). Due to the retrospective nature of this study, participants informed consent was exempted.

Among the enrolled patients with HER2-negative breast cancer, we compared the overall characteristics of the HER2-zero and HER2-low groups and analyzed the factors (including the HER2 status) influencing the pCR rate and DFS. We further conducted stratified research based on the HR status. We explore the characteristics and effects of the HER2 status separately within the HR-positive/HER2-negative breast cancer (HR +/HER2-BC) subgroup, also known as the Luminal subtype, and the HR-negative/HER2-negative breast cancer (HR–/HER2-BC) subgroup, also known as TNBC subtype. A study flow diagram is shown in Fig. 1.

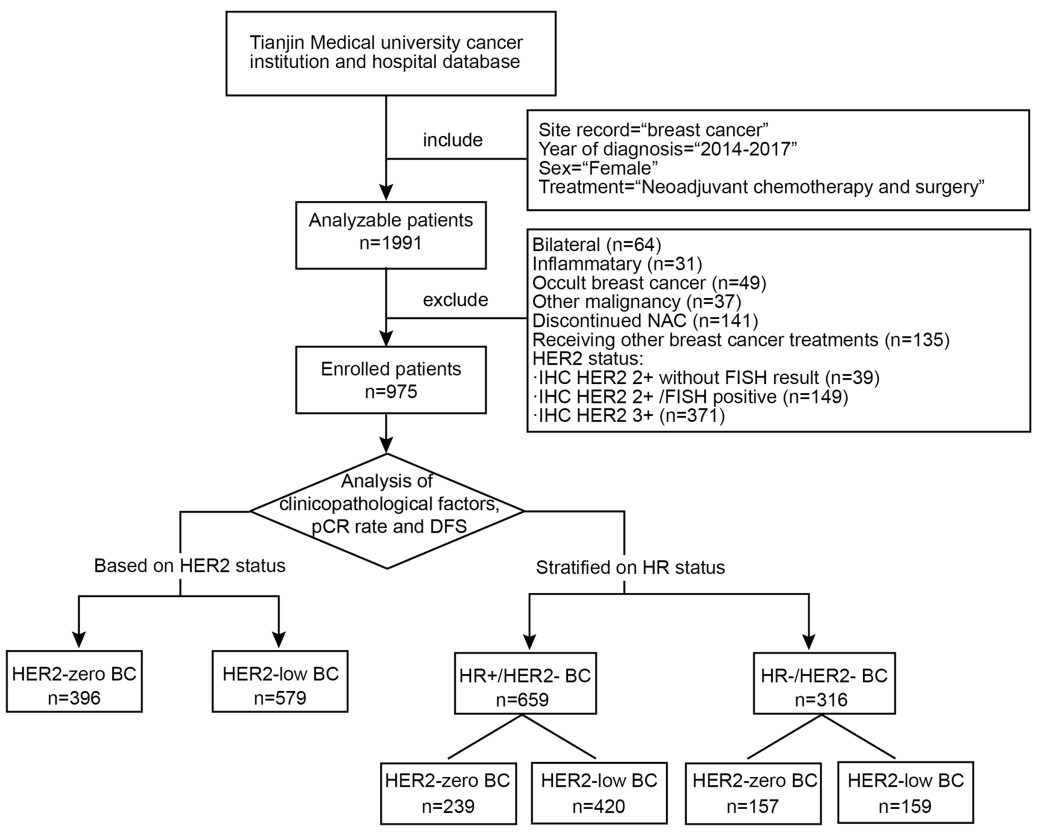

**Figure 1 Flow chart of the study design.** The flow chart shows the inclusion and exclusion criteria for breast cancer patients and the analysis workflow. NAC, neoadjuvant chemotherapy; HER2, human epidermal growth factor receptor 2; IHC, immunohistochemistry; FISH, fluorescence *in situ* hybridization; pCR, pathological complete response; DFS, disease-free survival; HR, hormone receptor.

## Diagnosis, IHC, and staging system

Immunostaining for ER, PR, and HER2 was performed according to the 2018 American Society of Clinical Oncology/College of American Pathologists guidelines (*Wolff et al., 2018*). ER and PR were deemed positive when the percentage of stained cells was ≥1%. ER and/or PR positive was defined as HR-positive, ER and PR negative was classified as HR-negative. Breast cancers with HER2 IHC scores of 0, 1+, or 2+ and negative FISH (IHC 2+/FISH−) are collectively referred to as HER2-negative. Among them, HER2 IHC 1+ or HER2 IHC 2+/FISH− is defined as HER2-low, and HER2 IHC 0 is defined as HER2-zero. According to the St. Gallen Guideline 2013, Ki-67 expression is classified as high or low based on a cutoff of 14% (*Goldhirsch et al., 2013*). HR and HER2 status in this study was determined by thick needle biopsy before NAC. Every semiquantitative scoring of immunostained section was independently analyzed by two pathologists. The sample was reassessed in case of discordance. Anatomical staging was performed according to the 8th edition of the American Joint Commission Cancer (AJCC) breast cancer staging system.

## Neoadjuvant treatment and efficacy evaluation

All patients received a minimum of two cycles of NAC regimens, including taxane and anthracycline, in the absence of neoadjuvant targeted or endocrine therapy. pCR was defined as the absence of invasive carcinoma in the breast or axillary lymph nodes after chemotherapy despite the possible presence of *in situ* residual ductal carcinoma components within the breast lesion (ypT0/is ypN0). All HR-positive patients received adjuvant endocrine therapy and regimens after surgery, including aromatase inhibitors (AIs) and selective estrogen receptor modulators (SERMs). All patients received standard chemotherapy after surgery in compliance with the national and international guidelines. Radiotherapy was required in cases of T3–T4 or nodal involvement before NAC.

## Follow-up

Patient follow-ups were conducted primarily through telephone inquiries, with information obtained from both inpatient and outpatient medical records. The follow-ups ended on November 30, 2022. A total of 99 cases were excluded in survival analysis due to missing data on survival state and survival time. DFS was defined as the time between the date of surgery and the date for which relapse or metastasize was confirmed or death from any cause.

## Statistical analysis

Data were analyzed using SPSS 25.0 software (IBM., Armonk, NY, USA). Count data are presented as composition ratios or rates. The chi-square test was applied to identify discrepancies in variable distributions. Binary logistic regression was used to identify factors influencing pCR. The Kaplan–Meier method was used to calculate survival rates and plot survival curves, and the Log-rank test was used to compare the survival differences between groups. The Cox proportional hazards regression model was used to analyze significant independent risk factors related to DFS for patients. $p$-value < 0.05 was considered statistical significance.

# RESULTS

## Clinicopathological factors associated with HER2-low and HER2-zero breast cancer

A total of eligible 975 patients with HER2-negative breast cancer between January 2014 and December 2017 were enrolled in this study. All 975 patients were all female, with a median age of 50 years (range: 24–76 years). Of these, 396 (40.6%) were HER2-zero, and 579 (59.4%) were HER2-low. The HER2-low group included 144 cases with HER2 IHC 2 +/FISH− (14.7%) and 435 cases with HER2 IHC 1+ (44.6%). The chi-square test showed that relative to patients with HER2-zero, those with low HER2 levels tended to be postmenopausal ($p$ = 0.016), had lower histological grades ($p$ < 0.001), and higher HR positivity ($p$ < 0.001). However, no significant differences were observed in age, primary tumor size (clinical T stage), regional lymph node metastasis (clinical N stage), pathological type, Ki-67 levels, type of NAC and adjuvant RT ($p$ > 0.05). Further binary

logistic regression analysis revealed associations between the HER2-low status and lower histological grade (OR = 2.47, 95% CI [1.32–4.61], $p$ = 0.004), and HR positive expression (OR = 5.22, 95% CI [3.65–7.46], $p$ < 0.001) (Table 1). We further stratified by HR status and compared the clinicopathological features of HER2-zero and HER2-low separately in the HR-positive and HR-negative subgroups. We found that in HR-positive breast cancer, HER2-low consisted of more patients with lower histological grade (grade I-II) than HER2-zero (77.1% $vs$. 29.7%, $p$ < 0.001). However, in the HR-negative group, the results were reversed (37.7% $vs$. 50.9%, $p$ = 0.002) (Table 2).

## Correlation between clinicopathological factors and pCR in HER2-negative breast cancer

The pCR rate for the 975 HER2-negative patients was 19.5% (190/975). The chi-square test indicated that pCR was more likely to occur in patients with HER2-zero, lower clinical T and N stages, higher histological grades, HR negative and higher Ki-67 level. Conversely, no significant associations were observed between pCR and age, menstrual status, and pathological type ($p$ > 0.05). Patients with HER2-zero status had a markedly greater pCR rate than those with HER2-low (24.0 % $vs$. 16.4%, $p$ = 0.003) (Table 3). Variables found to be significant ($p$ < 0.05) were included in the binary logistic regression analysis. This showed that clincal T and N stages, histological grade, HR status, and Ki-67 independently predicted pCR in patients with HER2-negative breast cancer. However, the HER2-low status was not an independent predictive factor for the pCR rate in multivariate analysis (Table 3 and Fig. 2).

## Effect of HER2 status on pCR rates across different stratified subgroups

Based on the previous analysis, it was observed that the HR positivity rate was significantly higher in the HER2-low than in the HER2-zero group. Therefore, we also explored the influence of HER2 status on pCR rate according to the stratification of HR status. In both the HR-positive and HR-negative breast cancer, T and N stages as well as histological grade were associated with the pCR rate (all $p$ < 0.05). In the HR-positive breast cancer, multivariate analysis revealed that both T and N stages as well as the HER2 status were independently influence the rate of pCR. Among them, HER2-low was identified as an inhibitory factor for the pCR rate of NAC (OR = 0.45, 95% CI [0.27–0.89], $p$ = 0.047). Specifically, HR-postive breast cancer with HER2-zero levels exhibited a significantly higher pCR rate than those with HER2-low (15.5% $vs$. 8.1%, $p$ = 0.003). However, no significant difference was observed in the pCR rates between HER2-zero and HER2-low cases in the HR-negative subgroup (36.9% $vs$. 38.4%, $p$ = 0.794) (Table 4 and Fig. 3).

Because HR is determined by both ER and PR, to further identify specific stratification factors, we also stratified according to ER and PR, respectively. Due to significant overlap in our cohort between groups stratified by HR and ER status, we observed similar results when stratifying by ER. In the ER-positive subgroup, the pCR rate of HER2-low was also significantly lower than that of HER2-zero (8.1% $vs$. 14.8%, $p$ = 0.008). In the multivariate analysis, HER2 status (OR = 0.58, 95% CI [0.30–0.95], $p$ = 0.048), along with T and N

**Table 1 Comparison of clinicopathological factors between patients with HER2-low and HER2-zero breast cancer.**

| Factors | n | HER2-zero (n = 396) | HER2-low | | Univariate analysis | Multivariate analysis | |
|---|---|---|---|---|---|---|---|
| | | | HER2-1+ (n = 435) | HER2-2+ (n = 144) | p-value | OR (95% CI) | p-value |
| Age | | | | | 0.059 | | |
| ≤40 | 194 | 87 (22.0%) | 92 (21.1%) | 15 (10.4%) | | | |
| 41–49 | 283 | 127 (32.1%) | 114 (26.2%) | 42 (29.2%) | | | |
| 50–74 | 492 | 179 (45.2%) | 227 (52.2%) | 86 (59.7%) | | | |
| ≥75 | 6 | 3 (0.7%) | 2 (0.5%) | 1 (0.7%) | | | |
| Menstrual status | | | | | 0.016 | | |
| Premenopausal | 430 | 193 (48.7%) | 181 (41.6%) | 56 (38.9%) | | Ref | |
| Postmenopausal | 545 | 203 (51.3%) | 254 (58.4%) | 88 (61.1%) | | 0.84 [0.59–1.20] | 0.342 |
| T stage | | | | | 0.226 | | |
| T1 | 162 | 70 (17.7%) | 79 (18.2%) | 13 (9.0%) | | | |
| T2 | 496 | 198 (50.0%) | 219 (50.3%) | 79 (54.9%) | | | |
| T3 | 240 | 107 (27.0%) | 90 (20.7%) | 43 (29.9%) | | | |
| T4 | 77 | 21 (5.3%) | 47 (10.8%) | 9 (6.2%) | | | |
| N stage | | | | | 0.939 | | |
| N0 | 371 | 153 (38.7%) | 167 (38.4%) | 51 (35.4%) | | | |
| N1-3 | 585 | 235 (59.3%) | 259 (59.5%) | 91 (63.2%) | | | |
| Unknown | 19 | 8 (2.0%) | 9 (2.1%) | 2 (1.4%) | | | |
| Pathological type | | | | | 0.954 | | |
| Invasive ductal carcinoma | 880 | 357 (90.2%) | 398 (91.5%) | 125 (86.8%) | | | |
| Invasive lobular carcinoma | 47 | 20 (5.0%) | 17 (3.9%) | 10 (6.9%) | | | |
| Others | 48 | 19 (4.8%) | 20 (4.6%) | 9 (6.3%) | | | |
| Invasive micropapillary carcinoma | 16 | 5 (1.2%) | 7 (1.6%) | 4 (2.8%) | | | |
| Metaplastic carcinoma | 8 | 4 (1.0%) | 2 (0.5%) | 2 (1.4%) | | | |
| Adenoid cystic carcinoma | 7 | 1 (0.3%) | 4 (1.0%) | 2 (1.4%) | | | |
| Apocrine gland cancer | 4 | 2 (0.5%) | 1 (0.2%) | 0 (0.0%) | | | |
| Mucinous carcinoma | 13 | 7 (1.8%) | 6 (1.3%) | 1 (0.7%) | | | |
| Histological grade | | | | | <0.001 | | |
| I | 111 | 23 (5.8%) | 53 (12.2%) | 35 (24.3%) | | Ref | |
| II | 424 | 128 (32.3%) | 219 (50.3%) | 77 (53.5%) | | 2.47 [1.32–4.61] | 0.004 |
| III | 415 | 240 (60.6%) | 143 (32.9%) | 32 (22.2%) | | 0.59 [0.33–1.04] | 0.066 |
| Unknown | 25 | 5 (1.3%) | 20 (4.6%) | 0 (0.0%) | | 1.63 [0.58–4.60] | 0.357 |
| HR status | | | | | <0.001 | | |
| Negative | 316 | 157 (39.6%) | 129 (29.7%) | 30 (20.8%) | | Ref | |
| Positive | 659 | 239 (60.4%) | 306 (70.3%) | 114 (79.2%) | | 5.22 [3.65–7.46] | <0.001 |
| Ki-67 | | | | | 0.564 | | |
| ≤14% | 128 | 49 (12.4%) | 49 (92.0%) | 30 (13.9%) | | | |
| >14% | 847 | 347 (87.6%) | 386 (8.0%) | 114 (86.1%) | | | |
| Type of NAC | | | | | 0.403 | | |
| Anthracycline | 718 | 297 (75.0%) | 338 (77.7%) | 83 (57.6%) | | | |

(Continued)

| Factors | n | HER2-zero (n = 396) | HER2-low | | Univariate analysis | Multivariate analysis | |
|---|---|---|---|---|---|---|---|
| | | | HER2-1+ (n = 435) | HER2-2+ (n = 144) | p-value | OR (95% CI) | p-value |
| Taxane | 43 | 21 (5.3%) | 9 (2.1%) | 13 (9.0%) | | | |
| Anthracycline and taxane | 182 | 66 (16.7%) | 77 (17.7%) | 39 (27.1%) | | | |
| Others | 32 | 12 (3.0%) | 11 (2.5%) | 9 (6.3%) | | | |
| Adjuvant RT | | | | | 0.213 | | |
| Yes | 855 | 348 (87.9%) | 378 (86.9%) | 129 (89.6%) | | | |
| No | 120 | 48 (12.1%) | 57 (13.1%) | 15 (10.4%) | | | |

**Note:**

HER2, human epidermal growth factor receptor 2; HR, hormone receptor; CI, confidence interval; NAC, neoadjuvant chemotherapy; RT, radiotherapy.

**Table 2 Clinicopathological factors comparison between HR-positive and HR-negative subgroups.**

| Factors | HR-positive (n = 659) | | p-value | HR-negative (n = 316) | | p-value |
|---|---|---|---|---|---|---|
| | HER2-zero (n = 239) | HER2-low (n = 420) | | HER2-zero (n = 157) | HER2-low (n = 159) | |
| Age | | | 0.055 | | | 0.447 |
| ≤40 | 53 (22.2%) | 75 (17.8%) | | 34 (21.7%) | 32 (20.1%) | |
| 41-49 | 80 (33.5%) | 112 (26.7%) | | 47 (29.9%) | 44 (27.7%) | |
| 50-74 | 105 (43.9%) | 230 (54.8%) | | 74 (47.1%) | 83 (52.2%) | |
| ≥75 | 1 (0.4%) | 3 (0.7%) | | 2 (1.3%) | 0 (0%) | |
| Menstrual status | | | 0.257 | | | 0.140 |
| Premenopausal | 102 (42.7%) | 159 (37.9%) | | 91 (58.0%) | 78 (49.1%) | |
| Postmenopausal | 137 (57.3%) | 261 (62.1%) | | 66 (42.0%) | 81 (50.9%) | |
| T stage | | | 0.732 | | | 0.355 |
| T1 | 31 (13.0%) | 46 (11.0%) | | 39 (24.8%) | 46 (28.9%) | |
| T2 | 122 (51.0%) | 217 (51.7%) | | 76 (48.4%) | 81 (50.9%) | |
| T3-4 | 86 (36.0%) | 157 (37.3%) | | 42 (26.8%) | 32 (20.2%) | |
| N stage | | | 0.367 | | | 0.073 |
| N0 | 79 (33.1%) | 162 (38.6%) | | 74 (47.1%) | 56 (35.2%) | |
| N1-3 | 154 (64.4%) | 248 (59.0%) | | 81 (51.6%) | 102 (64.2%) | |
| Unknown | 6 (2.5%) | 10 (2.4%) | | 2 (1.3%) | 1 (0.6%) | |
| Pathological type | | | 0.663 | | | 0.695 |
| Invasive ductal carcinoma | 210 (87.9%) | 378 (90.0%) | | 147 (93.6%) | 145 (91.2%) | |
| Invasive lobular carcinoma | 19 (7.9%) | 26 (6.2%) | | 1 (0.6%) | 1 (0.6%) | |
| Others | 10 (4.2%) | 16 (3.8%) | | 9 (5.7%) | 13 (8.2%) | |
| Invasive micropapillary carcinoma | 3 (1.3%) | 6 (1.4%) | | 2 (1.3%) | 5 (3.1%) | |
| Metaplastic carcinoma | 1 (0.4%) | 2 (0.5%) | | 3 (1.9%) | 2 (1.3%) | |
| Adenoid cystic carcinoma | 1 (0.4%) | 2 (0.5%) | | 0 (0.0%) | 4 (2.5%) | |
| Apocrine gland cancer | 0 (0.0%) | 1 (0.2%) | | 2 (1.3%) | 1 (0.6%) | |
| Mucinous carcinoma | 5 (2.1%) | 5 (1.2%) | | 2 (1.3%) | 1 (0.6%) | |
| Histological grade | | | <0.001 | | | 0.002 |

| Factors | HR-positive (n = 659) | | p-value | HR-negative (n = 316) | | p-value |
|---------|---------------------|-----------------|---------|---------------------|-----------------|---------|
| | HER2-zero (n = 239) | HER2-low (n = 420) | | HER2-zero (n = 157) | HER2-low (n = 159) | |
| I-II | 71 (29.7%) | 324 (77.1%) | | 80 (50.9%) | 60 (37.7%) | |
| III | 168 (70.3%) | 96 (22.9%) | | 72 (45.9%) | 79 (49.7%) | |
| Unknown | 0 (0%) | 0 (0%) | | 5 (3.2%) | 20 (12.6%) | |
| Ki-67 | | | 0.650 | | | 0.283 |
| ≤14% | 41 (17.2%) | 65 (15.5%) | | 8 (5.1%) | 14 (8.8%) | |
| >14% | 198 (82.8%) | 355 (84.5%) | | 149 (94.9%) | 145 (91.2%) | |
| Type of NAC | | | 0.430 | | | 0.023 |
| Anthracycline | 190 (79.5%) | 339 (80.7%) | | 107 (68.2%) | 82 (51.6%) | |
| Taxane | 11 (4.6%) | 10 (2.4%) | | 10 (6.4%) | 12 (7.5%) | |
| Anthracycline and taxane | 30 (12.6%) | 59 (14.0%) | | 36 (22.9%) | 57 (35.8%) | |
| Others | 8 (3.3%) | 12 (2.9%) | | 4 (2.5%) | 8 (5.1%) | |
| Adjuvant RT | | | 0.757 | | | 0.515 |
| Yes | 208 (87.0%) | 369 (87.9%) | | 140 (89.2%) | 138 (86.8%) | |
| No | 31 (13.0%) | 51 (12.1%) | | 17 (10.8%) | 21 (13.2%) | |

**Note:**
HER2, human epidermal growth factor receptor 2; HR, hormone receptor; NAC, neoadjuvant chemotherapy; RT, radiotherapy.

**Table 3 Correlation between factors and pCR rate among patients with HER2-negative breast cancer.**

| Factors | n | pCR (n = 190) | non-pCR (n = 785) | Univariate analysis p-value | Multivariate analysis OR (95% CI) | p-value |
|---------|---|---------------|-------------------|------------------------------|-----------------------------------|---------|
| Age | | | | 0.270 | | |
| ≤40 | 194 | 33 (17.0%) | 161 (83.0%) | | | |
| 41–49 | 283 | 51 (18.0%) | 232 (82.0%) | | | |
| 50–74 | 492 | 106 (21.5%) | 386 (78.5%) | | | |
| ≥75 | 6 | 0 (0.0%) | 6 (100.0%) | | | |
| Menstrual status | | | | 0.493 | | |
| Premenopausal | 430 | 88 (20.5%) | 342 (79.5%) | | | |
| Postmenopausal | 545 | 102 (18.7%) | 443 (81.3%) | | | |
| T stage | | | | <0.001 | | |
| T1 | 162 | 67 (41.4%) | 95 (58.6%) | | Ref | |
| T2 | 496 | 96 (19.4%) | 400 (80.6%) | | 0.48 [0.30–0.77] | 0.002 |
| T3 | 240 | 19 (7.9%) | 221 (92.1%) | | 0.22 [0.11–0.41] | <0.001 |
| T4 | 77 | 8 (10.4%) | 69 (89.6%) | | 0.27 [0.11–0.66] | 0.004 |
| N stage | | | | <0.001 | | |
| N0 | 371 | 118 (31.8%) | 253 (68.2%) | | Ref | |
| N1–3 | 585 | 69 (11.8%) | 516 (88.2%) | | 0.29 [0.19–0.42] | <0.001 |
| Unknown | 19 | 3 (15.8%) | 16 (84.2%) | | 0.28 [0.07–1.15] | 0.076 |
| Pathological type | | | | 0.059 | | |

(Continued)

| Table 3 (continued) | | | | | | |
|---|---|---|---|---|---|---|
| **Factors** | ***n*** | **pCR** **(*n* = 190)** | **non-pCR** **(*n* = 785)** | **Univariate analysis** **p-value** | **Multivariate analysis** **OR (95% CI)** | **p-value** |
| Invasive ductal carcinoma | 880 | 176 (20.0%) | 704 (80.0%) | | | |
| Invasive lobular carcinoma | 47 | 3 (6.4%) | 44 (93.6%) | | | |
| Others | 48 | 11 (22.9%) | 37 (77.1%) | | | |
| Histological grade | | | | <0.001 | | |
| I–II | 535 | 58 (10.8%) | 477 (89.2%) | | Ref | |
| III | 415 | 125 (30.1%) | 290 (69.9%) | | 3.38 [2.28–5.01] | <0.001 |
| Unknown | 25 | 7 (28.0%) | 18 (72.0%) | | 0.66 [0.24–1.83] | 0.422 |
| HR status | | | | <0.001 | | |
| Negative | 316 | 119 (37.7%) | 197 (62.3%) | | Ref | |
| Positive | 659 | 71 (10.8%) | 588 (89.2%) | | 0.22 [0.15–0.33] | <0.001 |
| Ki-67 | | | | 0.005 | | |
| ≤14% | 128 | 13 (10.2%) | 115 (89.8%) | | Ref | |
| >14% | 847 | 177 (20.9%) | 670 (79.1%) | | 2.00 [1.01–3.93] | 0.045 |
| HER2 status | | | | 0.003 | | |
| HER2-zero | 396 | 95 (24.0%) | 301 (76.0%) | | Ref | |
| HER2-low | 579 | 95 (16.4%) | 484 (83.6%) | | 1.00 [0.69–1.47] | 0.981 |

**Note:**
HER2, human epidermal growth factor receptor 2; HR, hormone receptor; pCR, pathological complete response. CI, confidence interval; OR, odds ratio.

stages, were identified as independent influencing factors of pCR rate. However, in the ER-negative subgroup, no significant difference was observed between the HER2-low and HER2-zero groups (38.1% *vs.* 37.7%, $p$ = 0.943) (Table S1). Stratified analysis by PR revealed no significant difference in the pCR rate between the HER2-low and HER2-zero groups, regardless of whether in the PR-positive subgroup (8.2% *vs.* 10.2%, $p$ = 0.423) or PR-negative subgroup (33.2% *vs.* 38.9%, $p$ = 0.240) (Table S2). These results suggest that stratification by ER status can more specifically differentiate the NAC efficacy between HER2-zero and HER2-low, while PR cannot.

## Factors influencing the pCR rate in HER2-low breast cancer

Univariate analyses of the 579 HER2-low cases indicated that clinical T stage ($p < 0.001$), clinical N stage ($p < 0.001$), HR status ($p < 0.001$), ER status ($p < 0.001$), and PR status ($p < 0.001$) had significant effects on pCR rate after NAC, whereas no statistically significant difference was found in age, menstrual status, pathological type, histological grade, and HER2 expression ($p > 0.05$). Binary logistic regression analysis verified that HR status and ER status independently predicted the pCR rate in patients with HER2-low breast cancer after NAC (Table 5).

## Prognosis of HER2-low and HER2-zero breast cancer

The median follow-up duration was 57 months (range 16–113 months). Overall, 99 (10.1%) patients were lost to follow-up, resulting in 876 patients with HER2-negative disease who had complete survival data. A total of 326 (37.2%) patients experienced a

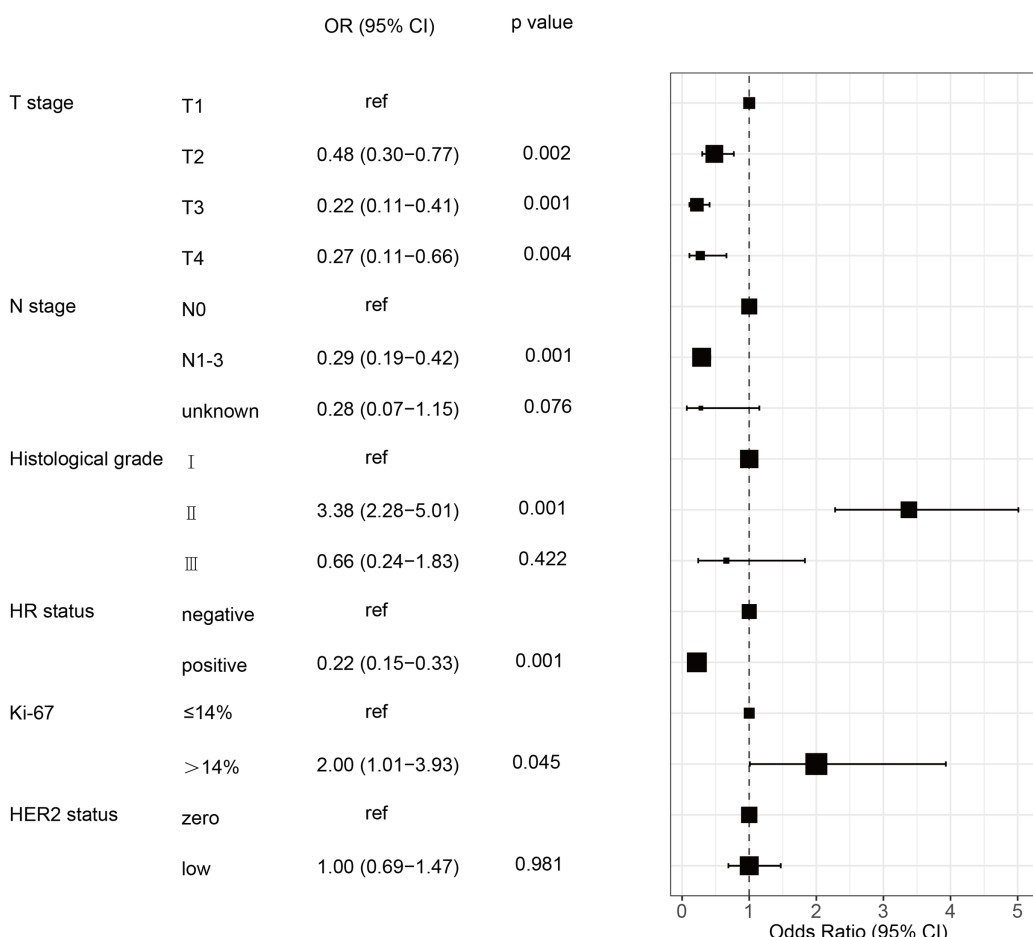

**Figure 2** **Forest plot of multivariate logistic regression analysis of risk factor associated with the pCR rate in HER2-negative breast cancer patients undergoing NAC.** HR, hormone receptor; HER2, human epidermal growth factor receptor 2; pCR, pathological complete response; OR, odd ratio; CI, confidence intervals; NAC, neoadjuvant chemotherapy.               

recurrence, metastasis, or death. Of these, 341 (38.9%) were in the HER2-zero group, and 535 (61.1%) were in the HER2-low group. The Kaplan–Meier curves showed that no significant difference in DFS between the HER2-zero and HER2-low status in the entire HER2-negative cohort ($p = 0.800$) as well as patients with the HR-positive ($p = 0.436$) and HR-negative ($p = 0.974$) (Figs. 4A– 4C, respectively). Significantly longer DFS were observed in patients who achieved pCR than in those who did not (all $p < 0.001$) (Fig. 5). However, no differences in DFS were observed between HER2-low and HER2-zero in the cohort who achieved ($p = 0.672$) or did not achieve pCR ($p = 0.370$) (Fig. 5A). Similarly, no differences in DFS were found between the HER2-low and HER2-zero groups among 300 (34.2%) patients with HR-negative breast cancer with pCR ($p = 0.979$) or among non-pCR cases ($p = 0.410$) (Fig. 5B). Moreover, DFS showed no differences in relation to HER2 status in 576 (65.8%) HR-positive breast cancer who achieved pCR ($p = 0.595$) or those who did not ($p = 0.292$) (Fig. 5C). In HER2-negative breast cancer, multivariate analysis found that T stage, N stage and pCR were independent influencing factors of DFS

Table 4 Factors associated with pCR rate in HR-positive and HR-negative subgroups.

| Factors | HR-positive | | | | | HR-negative | | | | |
|---|---|---|---|---|---|---|---|---|---|---|
| | n | pCR (n = 71) | Univariate analysis | Multivariate analysis | | n | pCR (n = 119) | Univariate analysis | Multivariate analysis | |
| | | | p-value | OR (95%CI) | p-value | | | p-value | OR (95%CI) | p-value |
| Age | | | 0.890 | | | | | 0.057 | | |
| ≤40 | 128 | 15 (11.7%) | | | | 66 | 18 (27.3%) | | | |
| 41–49 | 192 | 20 (10.4%) | | | | 91 | 31 (34.1%) | | | |
| 50–74 | 335 | 36 (10.7%) | | | | 157 | 70 (44.6%) | | | |
| ≥75 | 4 | 0 (0.0%) | | | | 2 | 0 (0.0%) | | | |
| Menstrual status | | | 0.940 | | | | | 0.280 | | |
| Premenopausal | 261 | 29 (11.1%) | | | | 169 | 59 (34.9%) | | | |
| Postmenopausal | 398 | 42 (10.6%) | | | | 147 | 60 (40.8%) | | | |
| T stage | | | <0.001 | | | | | <0.001 | | |
| T1 | 77 | 19 (24.7%) | | Ref | | 85 | 48 (56.5%) | | Ref | |
| T2 | 339 | 42 (12.4%) | | 0.42 [0.20–0.87] | 0.021 | 157 | 54 (34.4%) | | 0.37 [0.19–0.73] | 0.004 |
| T3 | 186 | 7 (3.8%) | | 0.12 [0.04–0.33] | <0.001 | 54 | 12 (22.2%) | | 0.23 [0.09–0.56] | 0.001 |
| T4 | 57 | 3 (5.3%) | | 0.17 [0.04–0.70] | 0.014 | 20 | 5 (25.0%) | | 0.35 [0.10–1.25] | 0.107 |
| N stage | | | <0.001 | | | | | 0.015 | | |
| N0 | 241 | 60 (24.9%) | | Ref | | 130 | 58 (44.6%) | | Ref | |
| N1-3 | 402 | 11 (2.7%) | | 0.08 [0.04–0.15] | <0.001 | 183 | 58 (31.7%) | | 0.67 [0.38–1.17] | 0.159 |
| Unknown | 16 | 0 (0.0%) | | – | – | 3 | 3 (100.0%) | | – | – |
| Pathological type | | | 0.890 | | | | | 0.183 | | |
| Invasive ductal carcinoma | 588 | 63 (10.7%) | | | | 292 | 113 (38.7%) | | | |
| Invasive lobular carcinoma | 45 | 3 (6.3%) | | | | 2 | 0 (0.0%) | | | |
| Others | 26 | 5 (19.2%) | | | | 22 | 6(21.4%) | | | |
| Histological grade | | | <0.001 | | | | | <0.001 | | |
| I | 87 | 9 (10.3%) | | Ref | | 24 | 10 (41.7%) | | Ref | |
| II | 308 | 19 (6.2%) | | 0.47 (0.19–1.18) | 0.108 | 116 | 20 (17.2%) | | 0.20 [0.07–0.54] | 0.002 |
| III | 264 | 43 (16.3%) | | 1.21 (0.47–3.12) | 0.696 | 151 | 82 (54.3%) | | 1.19 [0.46–3.04] | 0.722 |
| Unknown | 0 | 0 (0.0%) | | – | – | 25 | 7 (28.0%) | | 0.27 [0.07–1.03] | 0.055 |
| Ki-67 | | | 0.246 | | | | | 0.142 | | |
| ≤14% | 106 | 8 (7.5%) | | | | 22 | 5 (18.5%) | | | |
| >14% | 553 | 63 (11.4%) | | | | 294 | 114 (38.8%) | | | |
| HER2 status | | | 0.003 | | | | | 0.794 | | |
| Zero | 239 | 37 (15.5%) | | Ref | | 157 | 58 (36.9%) | | | |
| low | 420 | 34 (8.1%) | | 0.45 [0.27–0.89] | 0.047 | 159 | 61 (38.4%) | | | |

Note:
HER2, human epidermal growth factor receptor 2; HR, hormone receptor; pCR, pathological complete response; OR, odd ratio; CI, confidence intervals.

(Table 6). By stratification of HR status, we found that in HR-positive breast cancer, T stage, N stage and pCR remain to independently affect DFS. However, in HR-negative breast cancer, only pCR was an independent influencing factor for DFS (Table 7).
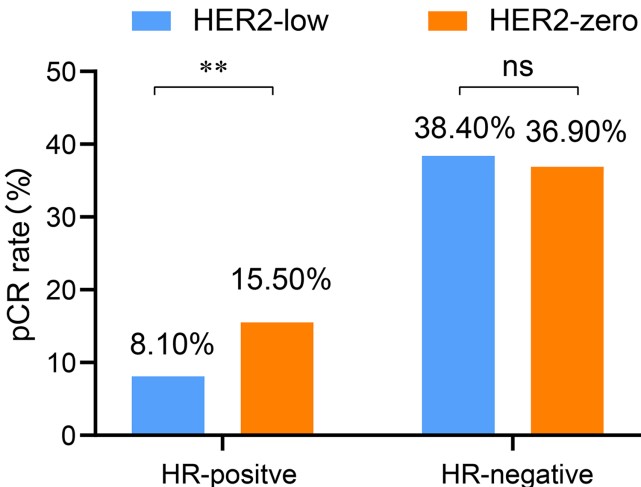

**Figure 3 Comparison of pCR rates between the HR-positive and HR-negative subgroups with low and zero HER2 status of HER2-negative breast cancer patients undergoing NAC.** HER2, human epidermal growth factor receptor 2; HR, hormone receptor; pCR, pathological complete response; NAC, neoadjuvant chemotherapy; ** $p < 0.01$; ns, not significant.

**Table 5 Analysis of factors associated with the pCR rate among patients with HER2-low breast cancer.**

| Factors | n | pCR n (%) | Univariate analysis p-value | Multivariate analysis OR (95% CI) | p-value |
|---|---|---|---|---|---|
| Age | | | 0.635 | | |
| ≤40 | 107 | 19 (17.8%) | | | |
| 41–49 | 156 | 21 (13.5%) | | | |
| 50–74 | 313 | 55 (17.6%) | | | |
| ≥75 | 3 | 0 (0.0%) | | | |
| Menstrual status | | | 0.163 | | |
| premenopausal | 237 | 45 (19.0%) | | | |
| postmenopausal | 342 | 50 (14.6%) | | | |
| T stage | | | <0.001 | | |
| T1 | 92 | 32 (34.8%) | | Ref | |
| T2 | 298 | 45 (15.1%) | | 0.53 [0.18–1.56] | 0.247 |
| T3 | 133 | 12 (9.0%) | | 0.88 [0.33–2.38] | 0.808 |
| T4 | 56 | 6 (10.7%) | | 0.27 [0.08–6.85] | 0.203 |
| N stage | | | <0.001 | | |
| N0 | 218 | 61 (28.0%) | | Ref | |
| N1–3 | 350 | 33 (9.4%) | | 0.23 [0.03–2.14] | 0.197 |
| Unknown | 11 | 1 (9.1%) | | 1.11 [0.11–5.74] | 0.927 |
| Pathological type | | | 0.385 | | |
| Invasive ductal carcinoma | 523 | 87 (16.6%) | | | |
| Invasive lobular carcinoma | 27 | 2 (7.4%) | | | |
| Others | 29 | 6 (20.7%) | | | |

(Continued)

| Factors | n | pCR n (%) | Univariate analysis | Multivariate analysis | |
| --- | --- | --- | --- | --- | --- |
| | | | p-value | OR (95% CI) | p-value |
| Histological grade | | | 0.075 | | |
| I | 88 | 19 (21.6%) | | | |
| II | 296 | 39 (13.2%) | | | |
| III | 175 | 31 (17.7%) | | | |
| Unknown | 20 | 6 (30.0%) | | | |
| HR status | | | <0.001 | | |
| Negative | 159 | 61 (38.4%) | | Ref | |
| Positive | 420 | 34 (8.1%) | | 0.02 [0.01–0.72) | <0.001 |
| ER status | | | <0.001 | | |
| Negative | 160 | 61 (38.1%) | | Ref | |
| Positive | 419 | 34 (8.1%) | | 0.02 [0.01–0.92] | <0.001 |
| PR status | | | <0.001 | | |
| Negative | 190 | 63 (33.2%) | | Ref | |
| Positive | 389 | 32 (8.2%) | | 0.32 [0.22–4.82] | 0.981 |
| HER2 status | | | 0.346 | | |
| IHC 1+ | 435 | 75 (17.2%) | | | |
| IHC 2+ | 144 | 20 (13.9%) | | | |

**Note:**
HER2, human epidermal growth factor receptor 2; HR, hormone receptor; pCR, pathological complete response; OR: odd ratio; CI: confidence intervals; ER, estrogen receptor; PR, progesterone receptor.

Our univariate and multivariate Cox regression analyses indicated that HER2 status was not a significant predictor of DFS in the entire HER2-negative cohort (Table 6), as well as in the HR-positive and HR-negative breast cancer (Table 7).

## DISCUSSION

Traditional anti-HER2 drugs, including trastuzumab, pertuzumab, and lapatinib, tend to be ineffective in patients with breast cancer who exhibit low HER2 levels (*Baselga et al., 2016*; *Burris et al., 2011*; *Kim et al., 2016*; *Krop et al., 2012*; *Lee et al., 2016*). Nevertheless, recently reported clinical trials have shown promise with new ADCs, such as DS-8201 (Trastuzumab Deruxtecan), in HER2-low patients (*Modi et al., 2020*). Thus, this has led to increased interest on the HER2-low expression in breast cancer. An investigation of 12,467 HER2-negative breast cancer cases in China showed that 54% of the patients were HER2-low (*Shui et al., 2020*). The sizable proportion of HER2-low breast cancer patients warrants thorough investigation. Here, 396 (40.6%), 144 (14.7%), and 435 (44.6%) patients had IHC 0, IHC 1+, and IHC 2+/FISH− HER2 levels, respectively, and with 59.3% of the HER2-negative patients classified as HER2-low, which was higher than the ratios found in previous studies.

Findings on the clinicopathological characteristics of patients with low HER2 expression breast cancer remain conflicting. Although patients with low HER2 expression are treated similarly to HER2-zero cases, the clinicopathological features and molecular

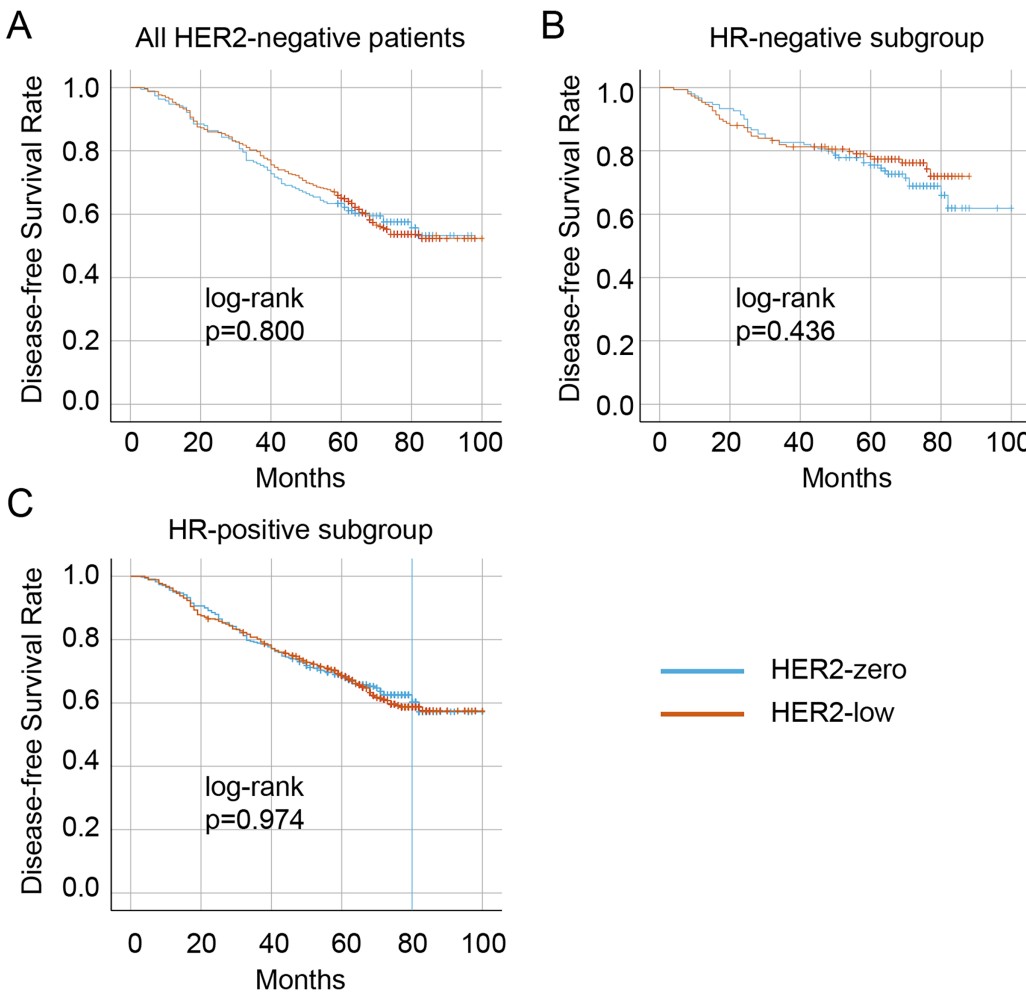

**Figure 4 Kaplan–Meier survival curves of DFS in HER2-low and HER2-zero breast cancer patients receiving NAC and subgroup survival curves.** Survival curves of DFS in the HER2-negative total population (A), the HR-negative subgroup (B), and the HR-positive subgroup (C) with low HER2 expression and zero HER2 expression DFS, disease-free survival; HER2, human epidermal growth factor receptor 2; HR, hormone receptor; NAC, neoadjuvant chemotherapy.

type of the two statuses differ considerably. A retrospective investigation of 3,689 HER2-negative patients revealed that HER2-low patients tended to be older, with higher T and N stages, histological grades, and proportion of HR positivity than HER2-zero cases (*Schettini et al., 2021*). Similarly, a prospective clinical trial of 2,310 patients found that low HER2 expression was associated with higher N stage and HR positivity, and lower histological grade (*Denkert et al., 2021*). On the other hand, a meta-analysis of 636,535 patients indicated that the proportion of the HER2-low was higher in HR-positive than HR-negative breast cancer (67.5% *vs.* 48.6%) (*Ergun, Ucar & Akagunduz, 2023*). Similarities with previous studies include our findings that HER2-low breast cancer was more prevalent in postmenopausal patients, those with lower histological grades, and those with HR positivity. Despite the absence of significant differences in T or N stage, our findings are consistent with the majority of studies and support a strong association

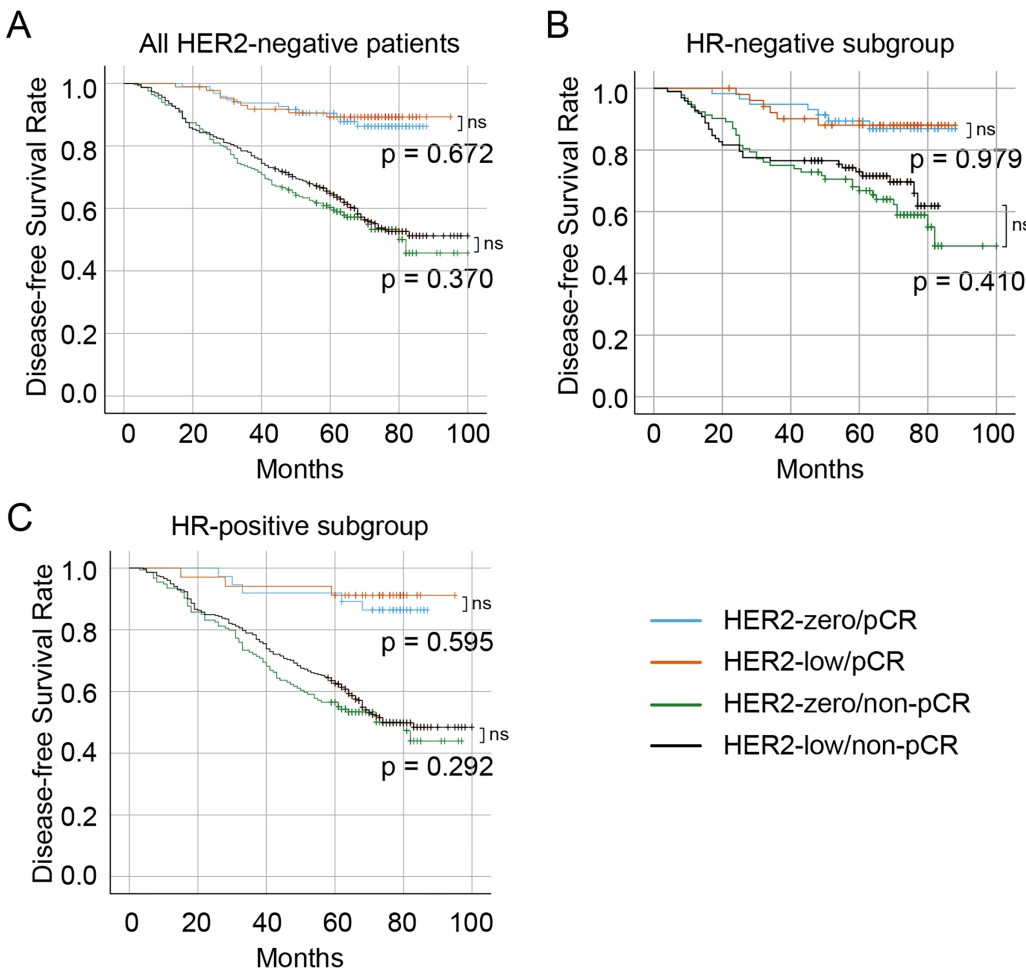

**Figure 5 Survival curves and subgroup survival curves of DFS in HER2-low and HER2-zero breast cancer patients undergoing NAC, stratified according to achievement of pCR.** Survival curves of DFS in the HER2-negative overall population (A), the HR-negative subgroup (B), and the HR-positive subgroup (C) with low HER2 expression and zero HER2 expression stratified according to the achievement of pCR. DFS, disease-free survival; HER2, human epidermal growth factor receptor 2; HR, hormone receptor; pCR, pathological complete response; NAC, neoadjuvant chemotherapy.

**Table 6 Univariate and multivariate analyses for DFS among patients with HER2-negative breast cancer.**

| Variable | Univariate analysis | | Multivariate analysis | |
|---|---|---|---|---|
| | HR (95% CI) | *p*-value | HR (95% CI) | *p*-value |
| Age | | | | |
| 41–49 | 0.81 [0.60–1.10] | 0.172 | | |
| 50–74 | 0.78 [0.59–1.03] | 0.082 | | |
| ≥75 | 1.34 [0.42–4.25] | 0.620 | | |
| Menstrual status | 1.05 [0.84–1.30] | 0.670 | | |

| Variable | Univariate analysis | | Multivariate analysis | |
|---|---|---|---|---|
| | HR (95% CI) | *p*-value | HR (95% CI) | *p*-value |
| T stage | | | | |
| T2 | 2.02 [1.34–3.03] | <0.001 | 1.52 [1.01–2.29] | 0.046 |
| T3 | 2.53 [1.65–3.87] | <0.001 | 1.63 [1.06–2.53] | 0.028 |
| T4 | 2.32 [1.39–3.89] | 0.001 | 1.55 [0.92–2.62] | 0.098 |
| N stage | | | | |
| N1–3 | 1.22 [1.06–1.56] | <0.001 | 1.13 [1.01–1.98] | 0.032 |
| Pathological type | | | | |
| Invasive lobular carcinoma | 1.35 [0.85–2.14] | 0.207 | | |
| Others | 0.93 [0.56–1.57] | 0.796 | | |
| Histological grade | | | | |
| II | 0.85 [0.59–1.22] | 0.381 | | |
| III | 0.81 [0.57–1.17] | 0.265 | | |
| Unknown | 1.43 [0.69–2.97] | 0.335 | | |
| HR[a] status | 0.69 [0.31–0.98] | 0.039 | 0.52 [0.07–3.88] | 0.523 |
| HER2 status | 1.03 [0.83–1.29] | 0.770 | | |
| Ki-67 | 0.85 [0.62–1.16] | 0.305 | | |
| pCR | 0.21 [0.14–0.33] | <0.001 | 0.26 [0.17–0.42] | <0.001 |

**Note:**
HER2, human epidermal growth factor receptor 2; HR[a], hormone receptor; DFS, disease-free survival; CI, confidence intervals; ER, estrogen receptor; PR, progesterone receptor; HR, hazard ratio.

**Table 7 Univariate and multivariate analyses for DFS among patients with HR-positive and HR-negative breast cancer.**

| Variable | HR[a]-positive | | | | HR[a]-negative | | | |
|---|---|---|---|---|---|---|---|---|
| | Univariate analysis | | Multivariate analysis | | Univariate analysis | | Multivariate analysis | |
| | HR (95% CI) | *p*-value | HR (95% CI) | *p*-value | HR (95% CI) | *p*-value | HR (95% CI) | *p*-value |
| Age | | | | | | | | |
| 41–49 | 0.70 [0.49–1.00] | 0.053 | | | 1.06 [0.58–1.93] | 0.853 | | |
| 50–74 | 0.77 [0.56–1.06] | 0.105 | | | 0.76 [0.43–1.34] | 0.338 | | |
| ≥75 | 0.37 [0.05–2.66] | 0.322 | | | 3.83 [0.64–5.49] | 0.258 | | |
| Menstrual status | 1.16 [0.90–1.49] | 0.255 | | | 0.72 [0.46–1.13] | 0.154 | | |
| T stage | | | | | | | | |
| T2 | 2.59 [1.40–4.81] | 0.002 | 1.94 [1.04–3.61] | 0.036 | 1.33 [0.75–2.37] | 0.328 | | |
| T3 | 3.10 [1.65–5.82] | <0.001 | 2.04 [1.08–3.87] | 0.028 | 1.59 [0.79–3.18] | 0.192 | | |
| T4 | 2.98 [1.47–6.03] | 0.002 | 1.98 [0.97–4.03] | 0.060 | 1.31 [0.48–3.59] | 0.595 | | |
| N stage | | | | | | | | |
| N1–3 | 3.21 [1.01–5.60] | 0.006 | 2.18 [1.03–4.89] | 0.015 | 2.36 [1.92–3.85] | 0.009 | 2.14 [0.65–3.47] | 0.076 |
| Pathological type | | | | | | | | |
| Invasive lobular carcinoma | 1.24 [0.78–1.99] | 0.364 | | | – | – | | |

(Continued)

| Table 7 *(continued)* | | | | | | | | |
|---|---|---|---|---|---|---|---|---|
| Variable | HR[a]-positive | | | | HR[a]-negative | | | |
| | Univariate analysis | | Multivariate analysis | | Univariate analysis | | Multivariate analysis | |
| | HR (95% CI) | *p*-value | HR (95% CI) | *p*-value | HR (95% CI) | *p*-value | HR (95% CI) | *p*-value |
| Others | 0.94 [0.48–1.83] | 0.858 | | | – | – | | |
| Histological grade | | | | | | | | |
| II | 0.72 [0.48–1.06] | 0.095 | | | 1.98 [0.70–5.57] | 0.196 | 1.37 [0.48–3.90] | 0.559 |
| III | 0.78 [0.53–1.16] | 0.216 | | | 1.45 [0.51–4.10] | 0.486 | 1.57 [0.55–4.45] | 0.400 |
| Unknown | – | – | | | 4.07 [1.25–13.32] | 0.020 | 2.49 [0.73–8.44] | 0.144 |
| HER2 status | 1.00 [0.77–1.31] | 0.973 | | | 0.83 [0.53–1.29] | 0.409 | | |
| Ki-67 | 1.02 [0.72–1.43] | 0.921 | | | 0.59 [0.29–1.24] | 0.164 | | |
| pCR | 0.17 [0.08–0.34] | <0.001 | 0.18 [0.09–0.37] | <0.001 | 0.29 [0.16–0.53] | <0.001 | 0.30 [0.16–0.57] | <0.001 |

**Note:**
HER2, human epidermal growth factor receptor 2; HR[a], hormone receptor; DFS, disease-free survival; CI, confidence intervals; ER, estrogen receptor; PR, progesterone receptor; HR, hazard ratio.

between HER2-low status and HR positivity in breast cancer (*Peiffer et al., 2023*; *Won et al., 2022*). In our study, the characteristics of HER2-low breast cancer overlapped with the reported features of HR-positive breast cancer (*e.g.*, lower grade, and prevalent among postmenopausal women). Therefore, we conducted stratified analyses based on the HR status. Notably, in HR-positive breast cancer, HER2-low tend to exhibit a lower histological grade. Conversely, in HR-negative breast cancer, HER2-low have a higher histological grade. This indicates that HER2-low status exhibits contrasting histological grade characteristic in different HR subtypes of breast cancer. HER2 and ER are key driving factors of cancer proliferation in breast cancer, and extensive preclinical research has been conducted to explore the crosstalk between the ER and HER2 signaling pathways (*Pegram, Jackisch & Johnston, 2023*). Despite being mildly expressed, HER2-low does not rule out complex interactions with ER, leading to distinct tumor biological features in different HR status.

In terms of NAC efficacy among patients with HER2-negative breast cancer in the Chinese population, our study found that the pCR rate of HER2-low was significantly lower than that of HER2-zero (16.4% *vs.* 24.0%, *p* = 0.003), but this was only limited to the univariate analysis. When both the HR and HER2 status were included in the multivariate analysis, the effect of HER2 was not significant (*p* = 0.880), but HR remained statistically significant (*p* < 0.001). This also suggests that the low expression of HER2 is likely to be influenced by the distribution of HR. Therefore, subgroup analysis of pCR rate was further conducted according to HR status. In HR-positive subgroup, the pCR rate of HER2-low group was still significantly lower than that of HER2-zero group (8.1% *vs.* 15.5%, *p* = 0.003), and further multivariate analysis showed that HER2-low was an independent predictive factor of pCR. In contrast, HER2 status did not affect the pCR rate in the HR-negative subgroup. *Peiffer et al. (2023)* analyzed data of 99,783 HER2-negative patients with pathological outcomes from the National Cancer Database, and similarly demonstrated a lower pCR rate in HER2-low compared to HER2-zero (16.3% *vs.* 23.6%,

$p < 0.001$). *Denkert et al. (2021)* prospectively studied 2,310 HER2-negative cases, and their results corroborated our findings, indicating a significantly lower pCR rate in HER2-low than in HER2-zero in the entire HER2-negative (29.2% *vs.* 39.0%, $p = 0.0002$), and HR-positive (17.5% *vs.* 23.6%, $p = 0.024$) patients. The difference of pCR rates between the HER2-zero and HER2-low was not statistically significant in the HR-negative subgroup (50.1% *vs.* 48.0%, $p = 0.21$). Meanwhile, *Ma et al. (2024)* analyzed a Chinese TNBC cohort comprising 1,445 cases, and revealed no significant difference in the pCR rates between HER2-low and HER2-zero (34.9% *vs.* 37.4%, $p = 0.549$). Furthermore, a recent meta-analysis involving 114,754 patients investigated the correlation between HER2 status and pCR rate, yielding conclusions consistent with ours. HER2-low was associated with a lower pCR rate, particularly in the HR-positive subgroup (*Molinelli et al., 2023*). These evidences corroborate our conclusion, indicating that the mild expression of HER2 (HER2-low) may inhibit the sensitivity to chemotherapy of the HR-positive subgroup, resulting in poorer efficacy of NAC in this subset.

As HR-positive is defined as ER and/or PR-positive, we also stratified patients with HER2-negative breast cancer according to ER and PR to determine more specific stratification factors. Since PR positivity is typically associated with ER positivity, ER-PR+ breast cancer patients are relatively rare, with only three cases in our cohort. Consequently, the results of ER stratification are generally consistent with those of HR stratification. Specifically, in ER-positive breast cancer, HER2-low has a lower pCR rate compared to HER2-zero, while no difference was observed in ER-negative breast cancer. However, when stratified by PR status, the pCR rates between HER2-low and HER2-zero were not different, regardless of PR negativity or positivity. Therefore, ER status may be a more specific stratification factor. In a study involving 234 patients of HER2-low and 91 patients of HER2-zero undergoing NAC, they observed relatively lower pCR rates of HER2-low than HER2-zero in both the HR-positive and ER-positive groups (9.0% *vs.* 13.5%; 7.9% *vs.* 11.8%, respectively). Despite statistical insignificance, this trend may be influenced by their relatively smaller sample size (*Zhou et al., 2023*).

Considering previous studies and our univariate and multivariate analyses in different stratifications of 975 HER2-negative breast cancer patients, we conclude that for HER2-negative cases undergoing NAC in the Chinese population, stratifying based on HR, particularly ER, can tailor more personalized treatment strategies for patients with varying HER2 statuses. Given the effectiveness of ADCs in HER2-low patients, we speculate that targeting HER2 therapy may significantly improve the pCR rates in the relatively chemotherapy-resistant subgroup of HR-positive/HER2-low or ER-positive/HER2-low.

However, several studies have revealed no significant differences in neoadjuvant treatment sensitivity based on HER2 status. A clinical study involving 855 HER2-negative patients found no significant differences in pCR rates following NAC based on HER2 status, whether in the HR-positive/luminal (13.0% *vs.* 9.5%, $p = 0.27$) or HR-negative/ TNBC cohort (51% *vs.* 47%, $p = 0.64$) (*de Moura Leite et al., 2021*). Similarly, a clinical study in France involving 511 patients found that HER2 status (low *vs.* zero) was not independently associated with pCR, either in HR-positive or HR-negative breast cancer (*Ilie et al., 2023*). While a recent analysis based on the National Cancer Database yielded

starkly different results, showing that regardless of HR status, HER2-low exhibited lower pCR rates than HER2-zero (*Li et al., 2024*). The results among different studies exhibit significant discrepancies. We speculate that this may be attributed to small sample sizes, ethnic differences, and subjectivity in HER2 testing in these studies and further research is needed with larger sample size and more balanced data in multicenter setting to compare the NAC efficacy between HER2-zero and HER2-low in breast cancer.

The impact of HER2-low status on prognosis is also highly contentious. A retrospective study on 1,433 patients with HER2-negative progressive disease who received treatment at the National Cancer Center of China revealed that HER2-low patients had longer overall survival (OS) than those with HER2-zero in entire cohort and HR-positive subgroup (*Li et al., 2021*). *Denkert et al. (2021)* reported similar findings, demonstrating that HER2-low patients had a higher 3-year DFS rate (83.4% *vs*. 76.1%, $p = 0.0084$) and 3-year OS (91.6% *vs*. 85.8%, $p = 0.0016$) than HER2-zero. In contrast, in the HR-negative subgroup, HER2-low demonstrated better prognosis, while no differences were observed in the HR-positive subgroup (*Denkert et al., 2021*).

However, *Zhang et al. (2022b)* reported different results, indicating that the DFS was similar in both HER2-low and HER2-zero cases ($p = 0.271$) and was not influenced by HR status, pCR, or molecular type. Similarly, *Ilie et al. (2023)* revealed that HER2 status (low *vs*. zero) did not affect recurrence-free survival (RFS) and OS. Additionally, other studies (*de Moura Leite et al., 2021*; *Xu et al., 2022*) found no differences in OS or DFS between patients with low and zero HER2 expression, irrespective of the HR status. These studies support our results, as we did not find any differences in DFS between HER2-low and HER2-zero in the entire population, stratified by HR status or pCR rates (with a median follow-up of 57 months). A recent survival data analysis of 987,934 cases from the national cancer database (with a median follow-up of 54 months) demonstrated that HER2-low only marginally improves prognosis (HR = 0.98 (0.97–0.99), $p < 0.001$) (*Peiffer et al., 2023*). Considering our findings on the distinct clinicopathological characteristics of HER2-low and the observed differences in sensitivity to NAC, we speculate that HER2-low may possess unique biological features, particularly in the HR-positive subgroup. However, this effect may be obscured by the efficacy of curative surgery, radiotherapy, and various systemic treatments, resulting in only a mild or negligible impact on long-term survival and recurrence. Notably, the current studies, including our own research, have a median follow-up period of approximately 5 years. Long-term monitoring is necessary to fully assess the prognosis of patients with HER2-negative breast cancer undergoing NAC.

## Limitations

This was a single-center retrospective study, necessitating further verification of its extrapolation. In addition, the interpretation of the results may have been affected by the heterochronous bias associated with the evolution and updating of pathological assessment technologies. The follow-up period was also relatively short, with only DFS used as an endpoint. Thus, large-scale sample studies with longer follow-up periods should be conducted to verify and refine these findings.

## CONCLUSIONS

In Chinese patients with breast cancer undergoing NAC, HER2-low exhibits distinct characteristics and pCR rate in different HR subgroups, with its diminished sensitivity to chemotherapy being particularly crucial in HR-positive breast cancer. Therefore, treatment should be tailored based on these subtypes. The impact of HER2 status on survival requires longer follow-up periods, and our study does not support its use for prognostic evaluation.

### Funding

This study was supported by research funding from the General Project of the Scientific Research Program of the Tianjin Municipal Education Commission (Grant no. 2020KJ136). The funders had no role in study design, data collection and analysis, decision to publish, or preparation of the manuscript.

### Grant Disclosures

The following grant information was disclosed by the authors:
General Project of the Scientific Research Program of the Tianjin Municipal Education Commission: 2020KJ136.

### Competing Interests

The authors declare that they have no competing interests.

### Author Contributions

- Shaorong Zhao conceived and designed the experiments, performed the experiments, analyzed the data, prepared figures and/or tables, authored or reviewed drafts of the article, and approved the final draft.
- Yuyun Wang conceived and designed the experiments, performed the experiments, analyzed the data, prepared figures and/or tables, authored or reviewed drafts of the article, and approved the final draft.
- Angxiao Zhou conceived and designed the experiments, performed the experiments, prepared figures and/or tables, authored or reviewed drafts of the article, and approved the final draft.
- Xu Liu conceived and designed the experiments, performed the experiments, analyzed the data, prepared figures and/or tables, authored or reviewed drafts of the article, and approved the final draft.
- Yi Zhang performed the experiments, prepared figures and/or tables, and approved the final draft.
- Jin Zhang conceived and designed the experiments, performed the experiments, prepared figures and/or tables, authored or reviewed drafts of the article, and approved the final draft.

## Human Ethics

The following information was supplied relating to ethical approvals (*i.e.*, approving body and any reference numbers):

The study conformed to the principles of the Declaration of Helsinki and was approved by Medical Ethics Committee of Tianjin Medical University Cancer Institute and Hospital (bc2023009).

## Data Availability

The raw data is available in the Supplemental File.

## Supplemental Information

Supplemental information for this article can be found online at http://dx.doi.org/10.7717/peerj.17492#supplemental-information.

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
