# Peer review of "Neoadjuvant chemotherapy efficacy and prognosis in HER2-low and HER2-zero breast cancer patients by HR status: a retrospective study in China"

_PeerJ, doi:10.7717/peerj.17492_

## Round 0.1 · original submission · Major Revisions

Authors are required to revise the manuscript and answer questions as suggested by the reviewers.

**Language Note:** The review process has identified that the English language must be improved. PeerJ can provide language editing services - please contact us at [email protected] for pricing (be sure to provide your manuscript number and title). Alternatively, you should make your own arrangements to improve the language quality and provide details in your response letter. – PeerJ Staff

Reviewer 1 ·

Basic reporting

The present study analysed the pathological complete response (pCR) and disease-free survival (DFS) in patients with human epidermal growth factor receptor 2 (HER2)-low and HER2-zero breast cancer who underwent neoadjuvant chemotherapy. Even though the study highlights the need for analysing the subgroups of HER2-low patients, specifically the HR positivity before treatment options, the findings don’t raise strong evidence in that regard. Since there are multiple research articles showing similar findings, the specific population group studied should be highlighted including in the title of the manuscript.
Here are some comments that could improve article:
1. Thorough proofreading of the manuscript is highly necessary for grammatical errors, repetition of words or sentences and typing errors.
2. The background in the abstract needs to be rewritten to include more details about the idea of the study. This should also include the population group (not only the hospital name). The specific details in the results section in the abstract can be reduced.
3. The expansion of all the short forms like HER2 should be added while mentioning the term for the first time and then using the short forms throughout the manuscript.
4. It would be helpful for readers from outside this study area if you could explain a little more about HER2, its role in breast cancer, how and why it is targeted, etc since the study is completely based on HER2.
5. Please detail in a couple of sentences on neoadjuvant therapy and the categories in the introduction.

Experimental design

No comment

Validity of the findings

No comment

Additional comments

Here are some additional comments:
1. The title needs to be reframed to include the population group that are studied.
2. Line 55: The authors are mentioning that HER2-negative means Luminal and TNBC. Does that mean HER2-0 and low are Luminal and TNBC? If so, did you study these groups of BC in the current manuscript? Please make it clear. Also, did you categorize the study subjects into these categories (Luminal A, B and TNBC)?
3. Results: In the first subtitle, please include few details of T and N stage and how Ki67 levels are determined, etc instead of directly mentioning the values.
4. Figures: The legend should include more details.
5. Figure 3: Please add the p-value in the graph itself as asterisk (*) and then mention the actual values in the legend.
6. Figure 5: Label the subgroups (A, B and C) in the graph itself and add the p-values too. Instead of red and orange colours, use some contrasting colors for easier analysis of the survival plot.
7. Please include more strong points supporting the findings of the study and the need for analysing HER2-low patients for HR positivity in the discussion and conclusion.
8. Please highlight the population group studied here in the discussion and conclusion also.

Reviewer 2 ·

Basic reporting

Major Comments:
This manuscript is trying to address the pathological complete response and disease-free survival in patients with HER2-low and HER2-zero breast cancer. They retrospectively studied data from 975 patients with HER2-low and HER2-zero breast cancer patients who underwent neoadjuant chemotherapy at the Tianjin Medical University Cancer Hospital between 2014 and 2017. The results show that HER2-low status exhibits different characteristics and chemotherapy sensitivity in the HR-positive and HR-negative subgroups from HER2-zero. However, HER2-low status is not a reliable factor for assessing long-term survival outcomes.

Experimental design

Replication of already available experimental design.

Validity of the findings

Validity of the findings need to emphasized in the conclusion part.

Additional comments

The study analyzed a large cohort of patients even though it is a replication study to confirm the already reported findings on the HER2-low and HER2-zero patients. Eventhough the study is not novel, it addressed the existing data in a new patient cohort from China. The authors have presented the data in a clear and professional language. I have the following comments to incorporate before acceptance.

1. Please explain the Line#68: How does the study help to diagnose the HER2-low BC patients?
2. The study should mention its significance/the relevance in the conclusion (compared to the given that there are a lot other similar studies).
3. Try to include some recent literature. For e.g., https://www.ncbi.nlm.nih.gov/pmc/articles/PMC9792954/

Minor comments

1. Line#114: Delete the repeated sentence, “The sample was reassessed in case of discordance.”
2. Be consistent with the terminology for HER2-0. The manuscript used both HER2-0 and HER2-zero.
3. Line#195: “Prognosis of HER2-low and HER2-low breast cancer”. Please correct this title.

Reviewer 3 ·

Basic reporting

1. Please be consistent with the terminology of HER2-negative/HER2-zero in abstract and the main text. At some points, did any patient undergo NGS testing to determine mutation status in HER2 or other relevant mutations?

2. Please pay attention to typo and grammar errors, for example, in line 190, “statically” should be statistically. The line 195, the subtitle is confusing, please consider change.

3. In line 197, does all the 876 patients have HER2-negative tumors? From my understanding, 396 (40.6%) in your cohort showed HER2-negative. In line 199, please explain further regarding the number discrepancy of HER2-zero groups throughout the whole manuscript.

Experimental design

1. The combination of ER and PR in the same subgroup of hormone receptor positive or negative should be discouraged and further analysis is needed to investigate the association of ER with HER2 low and HER2 negative, as well as the relationship of PR status in HER2 low and HER2 negative cases. The current conclusion is not specific, and the clinical application is limited in the setting of ER+/PR- or ER-/PR+ group.

2. In the HER-2 negative group, there are 23 cases of histologic grade I, what are the indications for these patients received neoadjuvant therapy?

3. In both table 1 and table 2, please provide detailed description of the “others” regarding the pathology type.

Validity of the findings

1. The rationale behind the exclude criteria is unclear, please explain the reason why patients with bilateral, inflammatory, or occult breast carcinoma, were excluded. It is recommended to have a diagram to illustrate how many cases are excluded and under which exclusion criteria to minimize bias.

2. It is quite interesting that the statistically significant conclusion of lower pCR rate in HER2-low group is only evident in univariate analysis but not the case when the hormone status is factored in. Again, this finding further suggests the necessity to study ER and PR positive individually rather than a group. Please explain further if there might be other confounding factors in your conclusion.

Additional comments

In study of Ma et.al 2023 (PMID: 37769330 ) of triple negative breast cancer, they concluded there are no difference of pCR between HER2-low and HER2-zero patients in triple negative cancer, please explain further how many cases in your study cohort are triple negative breast cancers (ER-, PR-, HER2-) and reconsider the analysis stratified by histology subtype.

---

## Round 0.2 · Minor Revisions

The authors are requested to carefully revise the manuscript and answer the questions raised by the reviewers.

Reviewer 1 ·

Basic reporting

The authors have addressed all the comments and corrected everything except one point. The authors were asked to add the specific population group in the manuscript title, abstract, etc, however, the authors have mistaken the comment. The 'population group' in the comments was referring to whether the population is a representation of the entire globe or whether all the patients from Tianjin Medical University Cancer Institute and Hospital are from a specific ethnic group. If they are from a specific ethnic group, the study cannot be concluded as a global study, hence the population group (here meaning the ethnicity) must be mentioned in the title and throughout the manuscript. Apart from that, all the comments were addressed by the authors. Thank you.

Experimental design

No comments

Validity of the findings

No comments

Additional comments

No comments

Reviewer 2 ·

Basic reporting

This manuscript is trying to address the pathological complete response and disease-free survival in patients with HER2-low and HER2-zero breast cancer. They retrospectively studied data from 975 patients with HER2-low and HER2-zero breast cancer patients who underwent neoadjuvant chemotherapy at the Tianjin Medical University Cancer Hospital between 2014 and 2017. The results show that HER2-low status exhibits different characteristics and chemotherapy sensitivity in the HR-positive and HR-negative subgroups from HER2-zero. However, HER2-low status is not a reliable factor for assessing long-term survival outcomes.

Experimental design

Addressed the comments.

Validity of the findings

Addressed well.

Additional comments

Addressed the comments.

Reviewer 3 ·

Basic reporting

No comment

Experimental design

No comment

Validity of the findings

No comment

Additional comments

The manuscript is significantly improved and the authors appropriately addressed all the concerns with detailed explanations.

---

## Round 0.3 · accepted · Accept

After revisions, all reviewers agreed to publish the manuscript. I also reviewed the manuscript and found no obvious risks to publication. Therefore, I also approved the publication of this manuscript.

Reviewer 1 ·

Basic reporting

I thank the authors for understanding the comments and making the necessary changes. Including the specific population group in the manuscript makes it more authentic and appropriate.

Experimental design

No comments

Validity of the findings

No comments

Additional comments

No comments